# Development and validation of the Family Resilience (FaRE) Questionnaire: an observational study in Italy

Flavia Faccio,[1,2] Sara Gandini,[3] Chiara Renzi,[2] Chiara Fioretti,[4] Chiara Crico,[1,2] Gabriella Pravettoni[1,2]

[1]Department of Oncology and Hemato-Oncology, Università degli Studi di Milano, Milan, Italy
[2]Applied Division for Cognitive and Psychological Science, European Institute of Oncology IRCCS, Milan, Italy
[3]Experimental Oncology, European Institute of Oncology IRCCS, Milan, Italy
[4]Department of Educational Sciences and Psychology, Università degli Studi di Firenze, Florence, Italy

**Correspondence to**
Flavia Faccio;
flavia.faccio.90@gmail.com

## ABSTRACT

**Objective** Develop and validate an instrument to assess family resilience and, more specifically, the family dynamics and resources, estimating the adaptation flexibility to cancer disease. Cohesion, communication, coping style and relational style were considered as critical functional areas in the construction of the instrument.

**Design** Two cross-sectional studies. Study 1: identification of factorial structure of the questionnaire in two samples with different cancer sites. Study 2: validation of the questionnaire in patients with cancer in two different phases of their therapeutic pathway.

**Participants and setting** A total of 213 patients with a histologically confirmed non-metastatic breast or prostate cancer and 209 caregivers were recruited for the two studies from an oncological hospital in Italy.

**Outcome measures** The Resilience Scale for Adults and the Family Resilience (FaRE) Questionnaire, developed by the researchers, were administered to all patients and caregivers who gave consent.

**Results** In study 1, the 60-item version of the FaRE Questionnaire underwent discriminant and construct validity, internal consistency and factorial analysis. Comparisons between patient and caregiver populations showed that patients perceived higher levels of family resources (p=0.048) and that patients with prostate cancer perceived less social support compared with patients with breast cancer (p=0.002). Factor analysis demonstrated four domains: communication and cohesion, perceived social support, perceived family coping, and religiousness and spirituality. In study 2, the validity and factorial structure of the final scale, composed of 24 items, were confirmed. The Cronbach alpha of all subscales was above 82. Normative values for patients with breast cancer can provide indications of family resilience levels.

**Conclusions** Preliminary findings showed acceptable psychometric properties for the FaRE Questionnaire to evaluate family resilience in oncological patients and their caregivers. Further research should test its sensibility to change to assess its use as a psychoemotional monitoring tool and its validity in other medical contexts.

## INTRODUCTION

Resilience is a multifaceted concept that can be described as the ability to mobilise resources and to adapt to challenging or adverse situations.[1 2] One of the internal

### Strengths and limitations of this study

► Questionnaire development and testing were conducted in two phases in order to assess the validity and reliability of the final version of the measure.
► This is the first measure of family resilience that adopts a systemic approach, considering both the patient and the caregiver, in light of increasing evidence on the vital role the caregiver.
► Results were controlled for the possible confounding sociodemographic variables.
► Only two oncological populations were recruited, making generalisability to all types of patients with tumour limited.
► Longitudinal measurements are advised to investigate test–retest reliability and responsiveness to changes in the measure.

resources of resilience is coping, which consists of a set of modifiable skills that can be learnt to deal with a stressful life event.[3] More and more often, resilience is studied as a relevant aspect when facing a disease such as cancer.[4–6] However, the impact of disrupting events, such as a cancer diagnosis, is not limited only to the individual; rather, it influences also their family and social network, which can initiate or support positive adaptation, as well as be overwhelmed by the demands of the disease and its treatments. According to the family systems theory,[7] a change or a perturbation occurring to one member of the system will affect also the other members, which may result in a smooth adaptation to a new homeostasis or in difficulties that prevent the readjustment process.[7] This translates into clinically significant levels of distress, higher risk of developing psychosocial problems, high levels of conflict and low family cohesion.[8] Going beyond the idea that families which encounter difficulties are 'damaged' or 'pathological',[9] resilience may represent a pivotal concept to analyse and support the family in moving towards a new

balance and in constructing the meaning of the cancer event.[10]

A substantial contribution to research in family resilience comes from two main models: the Resiliency Model of Family Adjustment and Adaptation[11] and the Family Resilience Framework,[9] both of which have been applied to oncological settings. Walsh's framework was chosen for various reasons: first, conceptualisation of family resilience in McCubbin *et al*'s model is quite rigid; it belongs to a first wave of resilience research which believed that one has to progress through specific steps in order to achieve resilience. In addition to this, due to the large number of variables involved in the model, it was possible for previous studies to validate only parts of it, never in its entirety.[12] Differently, Walsh[9 13] defines resilience as the family's ability to 'withstand and rebound from adversity, strengthened and more resourceful', including the capacity to cope and to adapt to the stressor and the concept of post-traumatic growth. She considers three major dimensions as contributing to the system's resilience: (1) belief systems, (2) organisational patterns and (3) communication processes. Each of these overarching constructs is composed of three subprocesses: making meaning of adversity, positive outlook, transcendence and spirituality, flexibility, connectedness, social and economic resources, communication/problem solving, clarity, open emotional expression and collaborative problem solving. These are considered mutually interactive and synergistic as they facilitate and sustain each other across systems and over time.[13] Walsh also proposed an integrative family resilience model for illness-related challenges, including cancer[14]; currently, there is no quantitative evidence regarding the framework's efficacy.

Only three questionnaires have attempted to capture the multiple dimensions of Walsh's[9] framework. Sixbey[15] developed the Family Resilience Assessment Scale (FRAS), a 66-item measure with responses given on a 4-point Likert scale. The statistical considerations and factor loadings showed that a nine-construct model of family resilience was not supported, and therefore, the number of factors was reduced to six[15]: (1) family communication and problem solving, (2) social and economic resources, (3) maintaining a positive outlook, (4) family connectedness, (5) family spirituality and (6) making meaning of adversity. One of the subscales, communication and problem solving, is an overarching construct of Walsh's model, which originally contained three subprocesses The questionnaire was administered to the general American population without considering an essential aspect of resilience, namely, whether an adversity had occurred. A second validation study suggested that the FRAS could be a reliable and valid measure in a population that was not experiencing an adversity.[16] This is the major weakness of the measure, which has then been administered to families that have experienced difficult diagnoses, such as autism spectrum disorder[17] and epilepsy,[18] where disagreement regarding the number of factors of the scale is highlighted.[18] In addition to this, the measure

was never administered to multiple members of a family; therefore, the perspective of the questionnaire is individualistic and is not representative of the family unit as a whole. As individual perceptions of which resilience resources are activated may vary, it is essential to compare the views of different family members.[19]

The other two measures developed according to Walsh's framework are the Family Resilience Assessment (FRA[20]) and the Walsh Family Resilience Questionnaire (Walsh[9]). The former is composed of 29 items on a 5-point scale response and was validated in a sample of women with a history of breast cancer. The developers[20] warned that the items did not always reflect Walsh's indicators, decided to group the items in the three overarching dimensions and suggested that socioeconomic resources have little value in breast cancer survivors. Lane and Meszaros imply that further studies should capture the pluralistic view of the family members.[20] To date, the FRA has not been used in other oncological or more general health contexts, suggesting the need for further studies in order to consider the instrument a valid measure. In addition to this, the recruited sample was small and different subgroups were present: survivors who had received diagnosis in the past 15 years, others in the past 5 years and another group of women undergoing treatment, of which half has a stage IV breast cancer diagnosis. Therefore, both current and retrospective accounts of resilience were collected. Finally, Walsh[19] developed her own self-report questionnaire, composed of 33 items that operationalise her nine key processes and provide a map of each family. While she states that her questionnaire is being applied in several international projects, data from these studies have not been published yet.

As there is only one measure of family resilience in patients with cancer,[15] which has an individualistic view of resilience and does not capture the systemic processes involved, a new measure including Sixbey's six factors was developed. These factors were chosen, compared with Lane's three factors, as, in our opinion, not all salient aspects of family resilience in cancer had been considered and to avoid construct under-representation, one of the major threats to construct validity.[21] The aim of this measure is to investigate family resources and strengths during cancer management, taking into account both the patient's and the caregiver's perspectives.

## MATERIALS AND METHODS
All study phases were presented in a single protocol to the local ethics committee, which approved all its parts on 11 April 2016. The aim of the first study was to develop a questionnaire, the Family Resilience (FaRE) Questionnaire to assess resilience in families affected by cancer. Following Messick's[21] guidelines for validating a measure, the following stages were conducted: substantive, structural and external. While substantive and structural steps were undertaken in study 1, the external stage was present in study 2.

## Study 1: item-generation phase and development of the questionnaire

For the substantive stage, a first version of the FaRE Questionnaire was constructed by the authors after a thorough revision of scientific literature in the field of resilience and cancer and after conducting informal, clinical appointments with 20 patients with cancer about strategies that helped them adjust and cope with the illness. The questionnaire was then developed and assessed by an expert panel composed of psychologists that conduct research in the field of resilience. Some minor adjustments to the questions were made, and five items were dropped as they were considered too similar to other statements. Finally, this preliminary version of the FaRE Questionnaire was administered to a pilot group of 10 citizens, 10 patients with cancer and 10 caregivers comparable to the target population in order to evaluate item comprehensibility, provide suggestions and indicate which items seemed unclear. Six items were reworded to improve understanding. The second version of the questionnaire was composed of 60 items referring to six aspects: (1) family communication and problem solving, (2) social and economic resources, (3) maintaining a positive outlook, (4) family connectedness, (5) family spirituality and (6) ability to make meaning from adversity. Questions were on a 7-point Likert scale (from strongly agree to strongly disagree).

## Item-selection phase

### Participants

Couples of patients and caregivers who fitted the inclusion criteria were asked to participate in the current study. Participation was proposed to a total of 146 couples, 42 of which refused due to little interest in the study or lack of time. A total of 105 patients (53 patients with breast cancer and 52 patients with prostate cancer) and 105 caregivers (ie, son/daughter, wife/husband/partner, father/mother and sister/brother) were recruited for the study at a comprehensive cancer centre in northern Italy. Patients were recruited from two outpatient clinics, one of the breast cancer division, the second of the urology division. In addition to this, patients undergoing oncological treatment were recruited from the radiotherapy division and day hospital. If compatible with the researcher's availability, consecutive patients who were waiting for their appointment were enrolled in the study. Once the study's aims and procedures were illustrated, all participants gave written informed consent.

Inclusion criteria for patients were the following: (1) between 25 and 80 years of age; (2) non-metastatic cancer diagnosis of breast or prostate cancer and (3) absence of psychiatric, addictive or cognitive disorders that would prevent compliance with protocol requirements.

The inclusion criteria for caregivers were (1) over 18 years old; (2) having a relative with a diagnosis of breast or prostate cancer and (3) absence of psychiatric, addictive or cognitive disorders that would prevent compliance with protocol requirements.

Due to incomplete answers of one patient, his data were not included in the database; therefore, the final sample was composed of 104 patients and 105 caregivers.

### Materials

The revised version of the FaRE Questionnaire, obtained at the end of the pilot study, was composed of 60 items and was administered to patients and caregivers participating in the study.

The Italian version of the Resilience Scale for Adults (RSA)[22] was chosen to assess the convergent validity of the FaRE Questionnaire. The RSA is a 33-item self-report scale for adults that measures six resilience factors: perception of self, planned future, social competence, structured style, family cohesion and social resources. It provides a total mean score, with higher scores indicating higher resilience. Four factors measure the individual's characteristics, one factor measures the family environment and the last factor measures social networks. Cronbach alphas for the Italian version ranged from 0.75 to 0.90 for all subscales and total score with the exception of structured style (0.34).

### Patient involvement and procedure

The patients in this study were participants. They were not involved in thedevelopment of the research question or in the recruitment phase. However, patients were asked for feedback on the readability of the questionnaire in the pilot phase, and their suggestions were incorporated in the final version of the FaRE Questionnaire.

Participants were approached by the researcher while they were in the waiting hall for the appointment with a healthcare professional (oncologist, surgeon or nurse). The researcher explained the purposes of the study and asked both the patient and the caregiver if they were interested in participating. If they answered affirmatively, an appointment was made after their medical consultation, during which further details were given, together with an informed consent that was signed by both the researcher and the participants. The questionnaires were administered on an iPad; for those who did not use electronic devices, paper-and-pencil versions of the questionnaires were given.

### Statistical analyses

All statistical analyses were conducted using SAS V.9.2. The first step was an exploratory factor analysis (EFA) to reduce the item pool.[23] The EFA was conducted using maximum likelihood factoring with orthomax rotation. Several criteria provided guidelines in selecting the number of factors to extract.[24] Factors with eigenvalues greater than 1 were retained, and an examination of scree plots of eigenvalues and total variance explained by retained factors helped to determine the total number of factors to retain (see figure 1). More generally, in the early stages and throughout the entire process, the items retained were those with the highest explained variance for each factor.

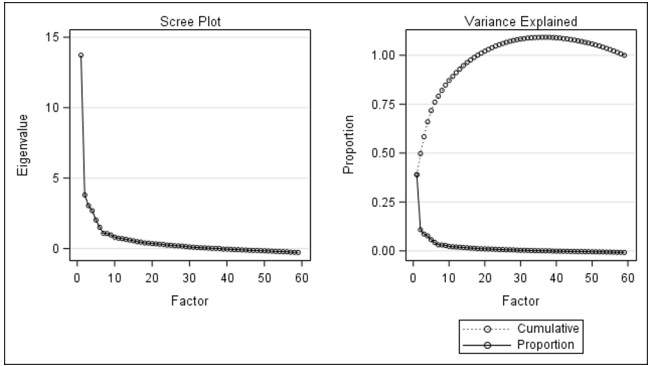

**Figure 1** Eingenvalues and proportion of variance of the factors.

The Kaiser-Meyer-Olkin (KMO) test for sampling adequacy was conducted on the sample to evaluate the adequacy of data.[25]

A Cronbach coefficient alpha was calculated for the entire scale and for each individual subscale to assure internal consistency and reliability. To further assess internal reliability, additional indices were computed: average variance extracted (AVE) values and Joreskog rho values.[26 27]

To evaluate the goodness of fit between the models and factorial invariance, we used fit indices, including the Bentler comparative fit index (CFI), the root mean square error of approximation (RMSEA), which indicates the amount of unexplained variance, and the standardised root mean square residual (SRMR).

Intraclass correlations (ICCs) were computed to measure inter-rater reliability.

Median values, SD, and first and third quartiles are presented for total and subscale FaRE scores. Values are presented also for patients and caregivers separately, by age groups, levels of education, marital status and number of children, and they are compared by Wilcoxon signed-rank tests. We did not expect differences between sociodemographic categories, while we expected that differences could arise between patients and caregivers, as activation of resources can differ, depending on the role of the family member in adjusting to the illness.

In order to investigate concordance between patients' and caregivers' scores, only scores of patients for which a corresponding caregiver's score was available were included. Differential scores were computed for each factor by subtracting the score of the patient from that of the corresponding caregiver. Paired t-tests were calculated to test the differences.

## Results

### Descriptive statistics

The median age of the population was 60 years old, with an IQR from 51 to 66.

Sixty per cent of caregivers were women. Seventy-one per cent of caregivers were wife/husband/partner, 14% were son/daughter, 11% were sister/brother and the remaining 4% were mother/father to the patient. The majority of

**Table 1** Sociodemographic characteristics of the sample

| | Total (n=209) | Patients (n=104) | CG (n=105) | P values |
|---|---|---|---|---|
| **Age (years)** | | | | |
| <50, n (%) | 43 (20.6) | 22 (21.2) | 21 (20) | 0.84 |
| ≥50, n (%) | 162 (77.5) | 80 (76.9) | 82 (78.1) | |
| Missing, n (%) | 4 (1.9) | 2 (1.9) | 2 (1.9) | |
| **Gender** | | | | |
| Male, n (%) | 93 (44.5) | 51 (49) | 42 (40) | 0.19 |
| Female, n (%) | 116 (55.5) | 53 (51) | 63 (60) | |
| **Education** | | | | |
| Elementary–middle, n (%) | 49 (23.4) | 23 (22.1) | 26 (24.8) | 0.76 |
| High school, n (%) | 90 (43.1) | 43 (41.3) | 47 (44.8) | |
| University, n (%) | 66 (31.6) | 35 (33.7) | 31 (29.5) | |
| Missing, n (%) | 4 (1.9) | 3 (2.9) | 1 (1) | |
| **Marital status** | | | | |
| Single,* n (%) | 35 (16.7) | 17 (16.3) | 18 (17.1) | 0.83 |
| Married, n (%) | 172 (82.3) | 87 (83.7) | 85 (81) | |
| Missing, n (%) | 2 (1) | 0 (0) | 2 (1.9) | |
| **Children** | | | | |
| None, n (%) | 39 (18.7) | 16 (15.4) | 23 (21.9) | 0.36 |
| 1–2, n (%) | 133 (63.6) | 68 (65.4) | 65 (61.9) | |
| >2, n (%) | 35 (16.7) | 20 (19.2) | 15 (14.3) | |
| Missing, n (%) | 2 (1) | 0 (0) | 2 (1.9) | |

*Single: never married or widow or separated.
CG, caregiver.

the participants was married or in a relationship (83.7% and 81% for patients and caregivers, respectively) and had children (84.6% and 76.2% for patients and caregivers, respectively). Most participants had a high school degree (41.3% for patients and 44% for caregivers) or higher (33.7% for participants and 29% for caregivers). No sociodemographic variables were significantly different between groups. Further descriptive data can be found in table 1.

Around 83% of patients with breast cancer and 45% of patients with prostate cancer were diagnosed with stages I and II, and the remaining patients had stages III and IV tumours. All patients with breast cancer and most patients with prostate cancer had undergone at least one type of oncological therapy previously (radiotherapy, chemotherapy and/or surgery). One patient with prostate cancer underwent positron emission tomography scan for tumour staging several weeks after completing the questionnaires, and the test indicated the presence of metastases. Further clinical data can be found in the online supplementary material.

### Factorial analysis

The KMO test scores for sampling adequacy were above 0.8 (overall KMO=0.92, patient sample KMO=0.9 and

caregiver sample KMO=0.84), indicating that the data were suitable for factorial analysis.[25]

Explained variance was over 10% for four factors; for the remaining two factors, the explained variance was under 5%, which was considered too low to be retained and were therefore excluded. Factor 5 had only two items, factor 6 had three items and in both factors, the items were unrelated to each other. EFA indicated four factors: one factor (factor 5) of the initial version of the questionnaire was removed ('ability to make meaning from adversity'), and some items of two factors, namely, 'family communication and problem solving' and 'family connectedness', were collapsed into a new subscale, communication and cohesion (supplementary material). For each factor, the items with the highest weights were selected. There are eight items for factors communication and cohesion and perceived social support, and there are four items for perceived family coping, religiousness and spirituality.

A sufficient model fit (Bentler CFI) was defined as 0.90 or greater, with an RMSEA of 0.06 or less.[28] The Bentler CFI was 0.94 ($\chi^2$=2571.16, df=276, p<0.001), with an RMSEA estimate 0.05, indicating an adequate fit between the model and the data.

### Final version of the questionnaire
The final factors of the FaRE Questionnaire are four (see online supplementary material). Communication and cohesion (items 1, 4, 7, 10, 13, 16, 19 and 22) refers to the family's openness in communicating about the illness and its impact on their daily life. It also involves shared problem solving and decision-making, resolution of conflict and openness regarding the range of feelings they are experiencing. The second factor, perceived social support (items 2, 5, 8, 11, 14, 17, 20 and 23), refers to mutual support, reaching out to extended kin and community for help with practical and emotional tasks. The third factor, perceived family coping (items 3, 9, 15 and 21), includes the ability to activate coping strategies and to develop inner strength to deal with illness

management. The concept of bouncing back and the ability to rebound from this stressful life event are present in this subscale. The last factor, religiousness and spirituality (items 6, 12, 18 and 24), refers to spiritual values, transcendent beliefs and congregational support.

### Internal consistency
An alpha of 0.70 is nowadays considered as acceptable for a new instrument.[29] The alpha computed for each of the four subscales exceeded the minimum value for a new tool. The Cronbach alphas for each factor were 0.88, 0.88, 0.82 and 0.86, respectively. AVE values were 0.52, 0.48, 0.58 and 0.71, and Joreskog rho values were 0.90, 0.88, 0.84 and 0.91, respectively. As an AVE value of 0.5 or more indicates good reliability and a rho value of 0.70 indicates acceptable reliability, it is possible to assume adequate internal consistency.[26 27]

### Comparison between patients and caregivers
All FaRE ICC coefficients were between 0.44 and 0.53, indicating an overall weak to moderate inter-rater reliability,[30] suggesting that there is a variation between patients and caregivers in rating the family resilience constructs.

Perceived family coping was significantly different between patients and caregivers, the former perceiving higher levels of family resources (median score of patients=6.3, median score of caregivers=6.0; p=0.048). A comparison was carried out also within patient–caregiver dyads, and the difference was not significant for any factor (see online supplementary material).

### Comparison between populations of patients with cancer
Perceived social support was significantly different between patients with breast cancer and patients with prostate cancer. The former reported receiving greater support from family and friends for their disease compared with the latter (p=0.002). Table 2 details the results of the non-parametric tests.

| Table 2 | Non-parametric tests on Family Resilience factors | | | | | | | | |
|---|---|---|---|---|---|---|---|---|---|
| | Patients (n=104) | | | | CG (n=105) | | | | |
| Factors | Median | SD | Q1 | Q3 | Median | SD | Q1 | Q3 | P values |
| Communication and cohesion | 6.6 | 0.9 | 5.7 | 7.5 | 6.5 | 0.8 | 5.7 | 7.3 | 0.238 |
| Perceived social support | 5.9 | 1.2 | 4.7 | 7.0 | 5.8 | 1.1 | 4.6 | 6.9 | 0.542 |
| Perceived family coping | 6.3 | 0.8 | 5.4 | 7.1 | 6.0 | 1.0 | 5.0 | 7.0 | 0.048 |
| Religiousness and Spirituality | 4.5 | 1.8 | 2.7 | 6.3 | 4.8 | 1.9 | 2.9 | 6.6 | 0.341 |
| | Breast (n=53) | | | | Prostate (n=51) | | | | |
| Communication and cohesion | 6.6 | 0.7 | 5.9 | 7.4 | 6.8 | 1.1 | 5.7 | 7.8 | 0.180 |
| Perceived social support | 6.3 | 0.8 | 5.4 | 7.1 | 5.5 | 1.3 | 4.2 | 6.8 | 0.002 |
| Perceived family coping | 6.0 | 0.9 | 5.1 | 6.9 | 6.5 | 0.7 | 5.8 | 7.2 | 0.171 |
| Religiousness and Spirituality | 4.8 | 1.8 | 3.0 | 6.5 | 4.3 | 1.9 | 2.3 | 6.2 | 0.285 |

P values are from Wilcoxon signed-rank tests. Q1 and Q3 are the first and third quartiles.
CG, caregiver.

## Study 2: validation of the FaRE Questionnaire

The second study was designed to validate the factorial structure of the questionnaire in a new sample of patients with breast cancer in two different phases of their therapeutic pathway: preadmission phase (either for day-surgery admission or for ordinary surgery admission) and treatment phase (surgery, radiation therapy or chemotherapy). In addition to this, the external stage of validation was conducted by assessing construct validity with some of the RSA and FaRE subscales. We expected to see positive correlations between (1) RSA and FaRE total scores, (2) RSA Family Cohesion Subscale and FaRE Communication and Cohesion Subscale, and (3) RSA Social Resources and FaRE perceived social support. While total scores should correlate as some individual resources could reflect in family resilience, the other two correlations were expected as they had common underpinning psychological constructs. Finally, significant correlations were hypothesised between RSA perception of self and FaRE family coping, as the RSA factor incorporates positive outlook, confidence and self-efficacy, aspects that are present in the systemic view of the family coping subscale.

### Participants

A total of 143 couples were approached to explain the study; 34 couples refused due to time constraints between appointments or due to little interest in participating. Only patients with breast cancer were recruited at this time due to easier access to patient waiting rooms, higher number of accesses in the hospital and higher compliance.

A total of 109 breast patients from a comprehensive cancer centre in northern Italy and 104 family caregivers were recruited for the study. Four caregivers were not able to come to the consultation on the scheduled day and thus could not provide informed consent; their responses therefore were not considered for the data analysis. The procedure of recruitment was the same as that for the first study.

### Materials

The final version of the FaRE Questionnaire obtained at the end of study 1, which consisted of 24 items, was administered to patients and caregivers (see online supplementary material). As in study 1, the RSA scale[22] was chosen to assess the convergent validity of the FaRE Questionnaire.

### Statistical analyses

SAS software V.9.2 was used to conduct all statistical analyses. Confirmative factorial analysis was conducted on the final version of the questionnaire. Multigroup factor analysis was conducted to test for factorial invariance across patients and caregivers. Wilcoxon signed-rank tests were conducted to evaluate possible differences in clinical and sociodemographic characteristics between the two samples and differences in factors between study 1 and study 2.

In order to assess construct validity of FaRE Questionnaire, we analysed whether the subscales of the questionnaire correlated with the RSA using Spearman correlations.

Finally, we tested the model separately in each of the group, caregivers and patients, and tested the goodness of fit of our four-factor model with a more conservative three-factor model to compare differences in practical fit indices and to evaluate whether strong factorial invariance, which requires a change in RMSEA of less than 0.015 or a change in CFI of less than 0.01,[28] is supported.

## Results

### Descriptive statistics

The median age of the population was 49, with IQR from 43 to 56. Thirty-three per cent of the caregivers were women. Most participants were married or in a relationship (80% and 74% for patients and caregivers, respectively) and had children (95% and 92% for patients and caregivers, respectively). The majority of the participants had a high school degree or higher (82% for patients and 82% for caregivers). More than half of the patients were not receiving oncological therapy at the time of recruitment and were diagnosed with stages I and II breast cancer. Further descriptive information can be found in table 3.

### Confirmatory factorial analysis

Weak factorial invariance suggested acceptable measurement invariance (patients, SRMR=0.079; caregivers, SRMR=0.080). The Bentler CFI was 0.90 and the RMR estimate was 0.07, indicating an adequate model of fit between the model and the data ($\chi^2$=3139.36, df=276, p<0.001). When testing the goodness of fit of our four-factor model with a more conservative, three-factor model, results yielded similar fit indices. The three-factor model had a Bentler CFI equal to 0.951 and an RMSEA estimate equal to 0.0516 ($\chi^2$=2044.46, df=190, p<0.0001), while the four-factor model's Bentler CFI was 0.942 and the RMSEA estimate was 0.0513 ($\chi^2$=2571.16, df=276, p<0.0001).

Difference between the two models in RMSEA scores was 0.00003 and in CFI was 0.009, providing evidence for strong factorial invariance.[31]

### Construct validity

Total scores of FaRE and RSA were positively correlated (ρs=0.43, p<0.0001). Factor communication and cohesion correlated positively with factor family cohesion in RSA (ρs=0.56, p<0.0001). Factor perceived social support correlated positively with factor social resources (ρs=0.54, p<0.0001). Factor perceived family coping correlated positively with factor structured style (ρs=0.30, p<0.0001) and with factor perception of self in RSA (ρs=0.42, p<0.0001).

### Differences between study 1 and study 2 samples

Wilcoxon signed-rank tests indicated no differences between the median values of factors in the samples in study 1 and study 2, both when considering together the

## Table 3 Descriptive statistics of the second sample

| | Overall (n=213) | Patients (n=109) | Caregivers (n=104) |
|---|---|---|---|
| **Age** | | | |
| <50, n (%) | 116 (54) | 109 (51) | 104 (49) |
| ≥50, n (%) | 97 (46) | 60 (55) | 56 (54) |
| **Children** | | | |
| 0, n (%) | 47 (22) | 49 (45) | 48 (46) |
| 1–2, n (%) | 153 (72) | 22 (20) | 25 (24) |
| >3, n (%) | 13 (6) | 82 (75) | 71 (68) |
| **Marital status** | | | |
| Single,* n (%) | 44 (21) | 20 (18) | 24 (23) |
| Married, n (%) | 164 (77) | 87 (80) | 77 (74) |
| Missing, n (%) | 5 (2) | 2 (2) | 3 (3) |
| **Educational level** | | | |
| Elementary–middle school, n (%) | 38 (18) | 20 (18) | 18 (17) |
| High school, n (%) | 92 (43) | 41 (38) | 51 (49) |
| University, n (%) | 82 (38) | 48 (44) | 34 (33) |
| Missing, n (%) | 1 (0) | 0 (0) | 1 (1) |
| **Stage** | | | |
| 0, n (%) | | 11 (10.1) | |
| I, n (%) | | 52 (47.7) | |
| II, n (%) | | 25 (22.9) | |
| III, n (%) | | 20 (18.8) | |
| **Previous therapy** | | | |
| Surgery, n (%) | | 42 (38.5) | |
| Chemotherapy, n (%) | | 19 (17.4) | |
| Hormone therapy, n (%) | | 2 (1.8) | |
| No therapy, n (%) | | 46 (42.2) | |
| **Ongoing therapy** | | | |
| Chemotherapy, n (%) | | 16 (14.7) | |
| Hormone therapy, n (%) | | 13 (11.9) | |
| Radiotherapy, n (%) | | 10 (9.2) | |
| No therapy, n (%) | | 70 (64.2) | |

*Single: never married or widow or separated.

patients and the caregivers, as well as when the patients and the caregivers were compared separately.

### Normative family resilience score

Due to disparity in number between patients with prostate cancer and patients with breast cancer, normative data were calculated on the total number of patients with breast cancer recruited for the two studies (n=317). The mean (5.7) and SD (0.78) for the total FaRE score were calculated. As with other clinical measures, scores between ±1 SD can be considered in the normal range, scores between −1 and −2 SD could indicate lower levels of family resilience and scores under 2 SD could be considered clinically relevant. For the last two ranges of scores, referral to support services would be recommended.

### DISCUSSION

The FaRE Questionnaire is designed to measure family resilience in oncological settings. Adopting a systemic approach, it considers both the patient and the caregiver's strengths and resources in managing the disease. Our initial version of the questionnaire had the same six family resilience constructs addressed by Sixbey[15]; however, a six-factor solution was not supported, and it was reduced to four factors. Compared with Sixbey's[15] questionnaire, the FaRE Questionnaire does not include 'making meaning of adversity' and 'positive outlook'; however, the latter is addressed in the perceived family coping subscale of the FaRE Questionnaire. Sixbey's connectedness construct is incorporated in family communication and cohesion. While Lane's[20] family resilience measure had highlighted that socioeconomic resources had little value to survivors of breast cancer, our measure suggests that social resources provide an invaluable support to the family coping with cancer and that economic resources were unrelated to the social ones. The development of the current questionnaire confirms that different resources are activated when families face cancer compared with other adversities and/or the general population.

An important aspect to keep in mind is the low interrater reliability between patients and caregivers, and this is partly explained by the significant differences found between some FaRE factors. Patients perceived significantly higher levels of family coping and higher levels of social support compared with their caregivers; although the difference in scores was small, it could be explained by the redirection and allocation of shared coping strategies towards the patient rather than towards the caregiver. This suggests that it is vital to consider the pluralistic view of family members as they are actively involved in cancer management. Further research should try and disentangle whether lower support and family coping scores of caregivers are of clinical significance, for instance, by assessing the association with mood profiles.

Recruitment of participants from eight different regions of Italy and the combination of cognitive interviews and expert review, which allowed us to determine the relevance of the items, are some of the strengths of the study. Another strength of the developed measure is its systemic view, which allows use in caregiver populations, for whom there are only a few questionnaires available, possibly because attention towards their challenges and needs has increased only in the last few years.[32 33]

This said, there were some limitations in the development of the current questionnaire. The sample size of the first study is relatively small, considering the number of items of the first version of the FaRE Questionnaire; a larger sample could have highlighted missing aspects.

Convergent and discriminant validity examining similarities and differences between the FaRE Questionnaire and other family resilience tools would strengthen the rigour of the questionnaire. In addition to this, the cut-off points of the total FaRE scores need validating in future studies in order to use them as screening for referral. In addition to this, it is important to evaluate whether the religiousness–spirituality subscale should be included in the total score of family resilience and whether it plays a key role in promoting resilience levels. To ensure generalisability of the results, a larger sample with diverse tumour diagnosis should be recruited, as our sample was composed of only gender-related tumours with a relatively high life expectancy; in addition to this, patients with other chronic and acute illnesses should be sampled to evaluate whether important family resilience aspects in disease management have been dropped. This instrument can be a useful tool to assess patients at baseline and for continued follow-up; for this reason, test–retest reliability and responsiveness to change should be conducted.

Preliminary evidence shows that the FaRE Questionnaire is a reliable and valid systemic measure of family resilience in oncological settings. The tool was designed to be implemented in a web platform as a part of the works of an European project, which aimed at promoting empowerment and self-management in patients with cancer.[34–36] As increasing attention is being paid to family resilience and its protective role in illness trajectories, once further validations are conducted, it can serve as a useful screening tool to identify patients and carers with few resources to deal with cancer-related issues. Family resilience screening would allow targeted interventions on development and promotion of specific strategies and resources that can help the family deal with the illness.

**Acknowledgements** We thank the patients for providing feedback and suggestions for improving the family resilience tool.

**Contributors** FF has made substantial contributions to the article conception and design, was involved in the recruitment of patients and in data entry and has given the final approval of the version to be published. SG has contributed to the analysis, interpretation of data and drafting of the work; provided a critical revision for important intellectual content and has given the final approval of the version to be published. CR has undertaken the literature review to construct the measure, has developed the items of the questionnaire, has been involved in drafting the article and has given the final approval of the version to be published. CF has undertaken the literature review to construct the measure, has developed the items of the questionnaire and has given the final approval of the version to be published. CC was involved in the recruitment of patients and data entry, and has given the final approval of the version to be published. GP has revised the article critically for important intellectual content and has given the final approval of the version to be published.

**Funding** This project has received funding from the European Union's Horizon 2020 research and innovation programme under grant agreement no. 643529.

**Disclaimer** This presentation reflects the authors' view. The commission is not responsible for any use that may be made of the information it contains.

**Competing interests** None declared.

**Patient consent for publication** Not required.

**Ethics approval** Obtained at the European Institute of Oncology.

**Provenance and peer review** Not commissioned; externally peer reviewed.

**Data sharing statement** Data leading to the results presented in this article are available upon reasonable request from Sara Gandini (sara.gandini@ieo.it).

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
