## [Reviewer comments · BMJ Open]

ARTICLE DETAILS

TITLE (PROVISIONAL)	Development and validation of a Family Resilience questionnaire (FaRE): an observational study in Italy
AUTHORS	Faccio, Flavia; Gandini, Sara; Renzi, Chiara; Fioretti, Chiara; Crico, Chiara; Pravettoni, Gabriella

VERSION 1 - REVIEW

REVIEWER	Sally Wheelwright University of Southampton, UK
REVIEW RETURNED	17-Jul-2018

GENERAL COMMENTS	Thank you for the opportunity to review this paper that describes the development and validation of a questionnaire to assess family resilience in families where one member has cancer. Below are my comments about each section. Abstract I think the conclusion has over-stepped the findings. Introduction 1. The introduction provides a reasonable summary of the concept of resilience. I think it could be strengthened by spelling out the difference between coping and resilience. 2. I would also suggest either editing the statement that "Although family resilience is a vital construct both in clinical and research settings..." or expand on this, as I do not think this has been demonstrated in the text. 3. "Sixbey developed a 66-item measure, the Family Resilience Assessment Scale (FRAS), on a 4-point Likert scale divided into 6 subscales.." Presumably you mean here that the FRAS is a 66-item measure with responses given on a 4-point Likert scale. 4. The description of Sixbey's study is a bit a confusing and perhaps the order of the information provided needs to be changed to fit what happened: did they develop a questionnaire based on 9 subscales which then became 6 subscales during development? Also what does validity mean in the context of the development study, given the criticism that it did not include participants who had suffered adversity? But then there is a description of its use in families facing adversity which begs the question why a new measure is needed. The explanation for this comes a couple of paragraphs later (it "has an individualistic view of resilience and does not capture the systemic processes
---

involved”). This should come in the critique of the FRAS and there should be a little more explanation about why this is a weakness.

5. Is there any other critique of the FRA beyond that it does not fit Walsh’s conceptualisation of family resilience?
6. Why is it important for a measure of family resilience to be disease specific? (paragraph 5). I’m also confused here that it states there is only one measure of family resilience in cancer (referencing the FRAS) but there is also the FRA.
7. The authors criticise Sixbey’s original development study but then follow this work by choosing, a priori to have six domains. Please justify this decision further.

Materials and method

1. The second sentence of the opening paragraph is a bit confusing and needs amending slightly to spell out simply that the aim of study one was to develop a questionnaire (the Family Resilience Evaluation Questionnaire) to assess resilience in families affected by cancer.
2. For Study 1, it would be helpful to have a couple of sentences about the overall plan – generate items through a literature review, carry out cognitive interviewing to check the items are comprehensible and then reduce the number of items.
3. In order to demonstrate the content validity of a questionnaire, you need to show that it is representative of the entire construct it is meant to be measuring. There needs to be something more convincing than simply saying, “a thorough revision of scientific literature in the field of resilience and cancer” was carried out.
4. The ‘Face validity’ section seems to me to describe more the process of cognitive interviewing to check comprehensibility: to check face validity you would need to check that people think the questionnaire is a measure of resilience. Please provide details of the ethics for this phase.
5. The use of the word “proposed” with respect to inviting participants to take part in research needs to be amended. Also do not use the word “drop-out” to participants who withdrew from the project.
6. What clinical information was collected about participants (section 1.2.1)? Where they recruited from outpatient clinics? Consecutive patients?
7. In section 1.2.2, it is stated that “The Italian version of the Resilience Scale for Adults (RSA) was chosen to assess the psychometric characteristics of the FaRE.” I think you mean the concurrent validity.
8. What was the procedure (for participants)?
9. What is meant by “cleanest” and most “interpretable” factor loading in section 1.2.4?
10. What were the hypotheses for discriminative and divergent validity (what differences were you expecting to find and why)?

Results

1. It would be good to see which items featured in which sub-scale in the supplementary material.
2. The issue for the section 1.2.5.5, relates back to point 10 above: why would you expect perceived social support to be different between breast and prostate patients, for example?
3. Edit “adopted” version of questionnaire.
4. I have a problem with the religiousness and spirituality sub-scale. I am not surprised these items hang together well psychometrically but what do they tell us about family resilience? They just tell us whether a family is religious/spiritual or not.

	5. Please provide some more details about the translation procedure and whether there were any cognitive interviews carried out with native English speakers. Some of the English is not very good e.g. items 1, 4. Study 2  1. This should follow the same overall structure as Study 1 (method, results) 2. I would expect a validation study to provide some evidence that the questionnaire is measuring what it is intended to measure, rather than just validating the factorial structure, which is what the opening sentence suggests. 3. Again, I would like to know a bit more about how the participants were recruited and what clinical data were collected. What was the procedure for participants? Also, I was expecting some English participants 4. Why would you expect any differences between patients and caregivers (section 2.3.4)? I don't understand why there is a difference between perceived social support in section 2.3.4 but no difference in section 2.3.5 5. I would lose section 2.3.7. These cut points need validating if they are going to be used as a screen for referral (and again, there is the issue of the religiousness and spirituality scale). Discussion  1. The opening paragraph is too long. I suggest summarising the findings and then separating the other issues into different paragraphs. I am not sure why the final sentence is included. 2. Much of the rest of the discussion will need to be adapted in light of the other comments I have made above. 3. Future research would also need to provide evidence of test-retest reliability and responsiveness to change.
--	--

REVIEWER	Inés Tomás University of Valencia Spain
REVIEW RETURNED	17-Jul-2018

GENERAL COMMENTS	This paper aims to develop and validate an instrument to assess family resilience. The authors make sound of their contribution stating that there is only one published measure of family resilience in cancer patients (the Family Resilience Assessment scale, FRAS; Sixbey, 2005) that has been adequately validated. According to the authors, this measure has an individualistic view of resilience and does not capture the systemic processes involved. Then, a new measure, taking into account both the cancer patient's perspective and the caregiver's perspective, is developed. The authors develop two independent studies and provide evidence of reliability and validity of the scale. The starting point seem promising. However, there are some important theoretical and empirical issues that should be addressed.  • In the introduction section (page 4) the authors talk about Froma Walsh's work as a substantial contribution to research in family resilience. They also expose three different questionnaires that attempt to capture the multidimensional model of Walsh. However, they do not mention or name the 9 dimensions that this model initially proposed. I propose to the authors to start mentioning the
---

9 initial dimensions proposed in Walsh's model. That would be useful in order to state an initial starting point, and later talk about which dimensions were kept or not in following studies or development of the questionnaires. Furthermore, no specific information about the FRA is offered (number of items, subscales...).

- When presenting the aims of the two different studies carried out, I propose to the authors to consider the process of validation, which involves different stages: substantive, structural and external (Messick, 1995). The substantive stage defines and delineates the construct under investigation. The structural stage pertains to establishing evidence of factorial validity and reliability relative to the construct of interest. The external stage examines whether the construct under investigation is related to other variables in accordance with the theoretical expectations. Then, they should reformulate the aim of the studies according to this framework, as the substantive stage seem not to be mentioned.

- The authors should state more clearly what are the contributions of the new created measure in regard to the previous one (the FRAS). It seems that the new measure has been created following the same 6 factor structure developed and validated for the FRAS. Why then create this new measure? What for? What are the specific contributions of the FaRe regarding the FRAS? Some point has been made by the authors regarding this issue, when they state that the FRAS has an individualistic view of resilience. However, could it just be solved by changing the initial instructions sentence? This point is not clear. More justification should be provided in order to justify to contribution of developing this new measure.

- Developing a questionnaire implies to accurately define the construct we want to measure, to deeply analyze and expose the model that we use to define the construct, and to discuss on the specific dimensions that configure the construct. None or poor development on that has been presented in the manuscript. In the introduction section I propose to the authors to further develop on that: what is resilience? What is family resilience? Which are the different models, theories or approaches on resilience/ family resilience? Which one has been selected and why? What does this model states about resilience/family resilience? Which are the dimensions that are considered in the model? What does each one mean or represent?

- In page 8, the authors indicate that "The Italian version of the Resilience Scale for Adults (RSA) was chosen to assess the psychometric characteristics of the FaRE". I suggest to be more concrete on that sentence and substitute "to assess the psychometric characteristics of the FaRE" for a more specific one (e.g., "to assess the convergent validity of the FaRE"; or "to provide evidence of validity of the FaRe based on the relationship with other variables" (following the current APA guidelines regarding validation of scales)).

- In page 8, a stablished scale (the Resilience Scale for Adults, RSA; Capanna et al., 2015) for measuring resilience is introduced to be used in the validation process of the FaRe. The authors indicate that this scale is composed of six resilience factors: four factors measure the individual's characteristics, one the family

environment, and the last one social networks. It seems that two of this dimensions (family environment and social network) represent family resilience. Again, more justification on the need of developing a new measure of family resilience, that do not overlap with previous published and well established scales, is needed.

- Nothing about data collection procedure is stated in the manuscript.

- In the Statistical analyses section, the authors rightly state that the first step was to carry out an EFA. I agree that the authors select the adequate statistical technique. However, the extraction method (principal axis factoring) and the rotation method (varimax rotation) selected to carry out the EFA are not the recommended ones.

It is also stated that “factors with eigenvalues greater than 1.0 were retained”. Although Kaiser’s rule is still the most widely used criteria to select the number of factors to retain, it is also the less recommended of all possible options.

I would suggest the authors to review the revised and updated guidelines to carry out an EFA (Lloret-Segura, Ferreres-Traver, Hernández-Baeza, & Tomás-Marco, 2014).

Moreover, there is no information about the statistical package used to carry out the analysis.

- Description of the sample is offered in the Results section. This information should appear in the Method section.

- Regarding the EFA results, no information on the adequacy of data to be factor analyzed is provided (e.g., Kaiser Meyer Olkin’s KMO test for sampling adequacy). Moreover, I would not completely rely on the results of the EFA, considering that the options to carry out this analysis were not the most adequate ones (as previously stated). Furthermore, the scree plot that appears in page 24 seem to indicate a six factor solution as a plausible one. More information on the criteria used to take decisions should be offered. Why the “Ability to make meaning from adversity” factor was removed?

Regarding the information provided of goodness of fit indices, it is not clear whether this fit correspond to the initial model with the complete pool of items, or to the model with the removed items. Moreover, information of the number of items that were developed for the initial 6 dimensions of the questionnaire, and number of items removed should be offered....

- I have a concern regarding the final questionnaire composition. The initial pool of items was 60, and before the EFA, just 24 were retained. This item deletion could have affected the construct representation? That is, some specific and valuable aspects could now not be represented by the items remaining in the questionnaire? Moreover, 6 dimensions were proposed initially, and finally just 4 were kept. This dimension dropping could be affecting the construct validity? That is, could be that some relevant dimension for family resilience is now missing?

- To further assess the internal reliability of the scales, the authors should also provide the Average Variance Extracted (AVE) values and composite reliability values (ρ).

- Authors should pay attention to which pieces of information appear in which section. For example, include in the Analyses section all the information related to the analyses carried out, and leave the Results section just to comment on specific results. As an example, the following text figures in the Results section, when it should be placed in the Analyses section : “In order to assess construct validity of FaRE, we analyzed whether the subscales of the questionnaire correlated with the RSA using Spearman correlations”.
- Authors are also encouraged to address the validation process within the framework of the AERA, APA and NCME (2014) guidelines. It affects the expressions and terms used, as it is recommended to refer to evidences of validity and the different sources of evidence of validity. [American Educational Research Association, American Psychological Association, & National Council on Measurement in Education. (1999). Standards for educational and psychological testing. American Educational Research Association.]
- In order to provide evidence of validity of test scores of the FaRe based on the relationship with other variables, as previously indicated, the authors also collect measures with the Resilience Scale for Adults (RSA). However, it should be advised which kind of relationships are expected in order to confirm that the results provide evidence of validity. Which kind of pattern relationship are expected? Which dimensions should show higher relationships according to their content and meaning? Are these expected pattern in the correlations supported? We also encourage the authors to present the correlations in a table for better understanding), and justify why they offer Spearman correlations instead of Pearson correlations.
- I cannot see the point of the results that appear in the “1.2.5.5. Discriminant validity” section. In which way the reported results provide evidence of discriminant validity? Are there any previous theoretical or empirical foundations to expect that the values in the dimensions of the FaRe scale should be different among patients and caregivers, or among breast and prostate cancer patients?
- In the second study, no information on the kind of factorial analysis that was carried out is reported. Was a Confirmatory Factor Analysis (CFA)? Following the recommendation on scale validation, in this step, a CFA should be used in order to confirm the factorial structure that appeared in the EFA carried out in Study 1. Again, no information on the statistical software used is provided.
- In the Discussion section it is stated: “The questionnaire confirms Sixbey’s assumption that the nine-factor family resilience model developed by Walsh can be reduced”. To state that, the FaRe questionnaire developed in this study should have considered the initial 9 factor proposed in Walsh’s model. However, the starting point was a 6 factor model.
- In the Discussion section authors also state: “Lane’s family resilience measure had highlighted that social and economic resources had little value to breast cancer survivors and our measure confirms this”. From which part of the results section can this conclusion be drawn?

	 • Most of the discussion section focus on the comparison between breast and prostate cancer patients scores, and between patients and care givers scores. None of this comparisons are in the aim of the study.
--	---

REVIEWER	Erik Farin-Glattacker Section of Health Care Research and Rehabilitation Research, Faculty of Medicine and Medical Center - University of Freiburg, Germany
REVIEW RETURNED	20-Jul-2018

GENERAL COMMENTS	The manuscript reports on the development and validation of a family resilience questionnaire. This is a relevant research topic and the study seems carefully conducted, but there are some methodological vagueness and weaknesses. My major concern is about the handling of patient and caregiver data. It seems as if these data were combined to conduct psychometric testing with the whole data set. As the perspectives of patients and caregivers concerning adaptation flexibility to the disease of the patient may be quite different, this approach seems disputable. I would advise to test first factorial invariance of the FaRE instrument in patients and caregivers. There are at least two levels of factorial variance that may be of interest in this case: „weak factorial invariance” across patients and caregivers and “strong factorial invariance”. Comparing mean values of patients and caregivers in a methodological paper without testing factorial invariance across these groups may lead to false conclusions. Concerning chapter 1.2.5.5.1 Comparison between patients and caregivers: As patients and caregivers are approached as raters of the same concept (family resilience) I would advise to compute intraclass correlations (ICC) as estimates of interrater reliability. Testing construct validity demands clear hypotheses about the expected correlations (e.g., COSMIN checklist manual). These hypotheses should be stated and justified. In the current version of the manuscript this is missing. Factors that were considered in construct validity testing differed in the methods section (chap. 1.2.4) and in the results section (chap. 1.2.5.4). The kind of the relation between patient and caregiver (e.g., son/daughter, wife/husband) was ignored. At least the authors should give this information in the table depicting characteristics of the sample. The criteria that lead the authors to extract 4 factors in the factor analysis should be stated more clearly. In my view the scree plot does not support this. Were aspects of interpretability decisive? As a rule of thumb, 5 participants per item are necessary to conduct exploratory factor analyses (60 items * 5 participants = 300). In view of this, the sample size of the two sub-studies is too small. At least, this point should be discussed as limitation.
---

	The inclusion criteria pose some questions: What is the reason for different age inclusion criteria for patients (≥ 25, ≤ 80) and caregivers (≥ 18)? Page 14: „The Bentler Comparative Fit Index was 0.88, the RMR Estimate 0.07, indicating a good model of fit between the model and the data.“ I can't follow this. On page 10 the authors state that CFI should be greater than 0.90. Page 16: „The FaRE questionnaire is designed to measure family resilience in medical settings.“ This assertion seems to be overdrawn as the authors investigate only cancer patients. The discussion should include a more thorough consideration of limitations. The strengths of the study mentioned in the discussion section („The sample size, the use of a standardised measure to compare factors, the goodness of fit between model and data, and the high internal consistency are some of the strengths of the study.“) should be reconsidered. The sample size is quite small given the statistical analyses that were conducted. Goodness of fit values and internal consistency values are results, not strengths.
--	---

VERSION 1 – AUTHOR RESPONSE

Reviewer(s)' Comments to Author:

Reviewer: 1

Reviewer Name: Sally Wheelwright

Institution and Country: University of Southampton, UK

Please state any competing interests or state 'None declared': None declared

Please leave your comments for the authors below

Thank you for the opportunity to review this paper that describes the development and validation of a questionnaire to assess family resilience in families where one member has cancer. Below are my comments about each section.

Abstract

I think the conclusion has over-stepped the findings.

We agree with this observation and have modified the Conclusion of the abstract accordingly.

Introduction

1. The introduction provides a reasonable summary of the concept of resilience. I think it could be strengthened by spelling out the difference between coping and resilience.

Resilience is a complex, multifaceted process that helps individuals and families to bounce back and adapt to the stressful life event. Coping mechanisms and behaviours are only one of the protective resources of resilience. Therefore, resilience is made up of many factors, including coping. This has now been addressed in the Introduction's first page, line 2.

2. I would also suggest either editing the statement that “Although family resilience is a vital construct both in clinical and research settings...” or expand on this, as I do not think this has been demonstrated in the text.

We agree with the reviewer and have removed the statement as expanding it would have made the Introduction too long, considering also Reviewer 2’s comments on restructuring and expanding on the Introduction.

3. “Sixbey developed a 66-item measure, the Family Resilience Assessment Scale (FRAS), on a 4-point Likert scale divided into 6 subscales..” Presumably you mean here that the FRAS is a 66-item measure with responses given on a 4-point Likert scale.

Yes, we have modified the text accordingly.

4. The description of Sixbey’s study is a bit a confusing and perhaps the order of the information provided needs to be changed to fit what happened: did they develop a questionnaire based on 9 subscales which then became 6 subscales during development? Also what does validity mean in the context of the development study, given the criticism that it did not include participants who had suffered adversity? But then there is a description of its use in families facing adversity which begs the question why a new measure is needed. The explanation for this comes a couple of paragraphs later (it “has an individualistic view of resilience and does not capture the systemic processes involved”). This should come in the critique of the FRAS and there should be a little more explanation about why this is a weakness.

We agree and have included a more detailed explanation of the critique of the FRAS when we describe the measure. We also agree with the reviewer regarding the validity of the FRAS. Sixbey argues that her measure is a valid and reliable one, however as it was administered to the general population which has not necessarily suffered an adversity, it is difficult to consider it a valid measure. In addition to this, within the selected sample there might be individuals suffering from a stressful life event and this may have altered the results and the final item selection.

5. Is there any other critique of the FRA beyond that it does not fit Walsh’s conceptualisation of family resilience?

The sample size for the two phases of measure development (pilot version and final version) was small, 44 and 113 respectively. In addition to this, while the authors acknowledge the importance of a pluralistic view of family resilience they decide to adopt an individual perspective. Finally, the sample was composed of different subsamples: most of the women were breast cancer survivors who had received a diagnosis in the past 5 or 15 years, a small number of women were still receiving treatment (19), eight of which had just received a stage IV breast cancer diagnosis. It seems clear that most of the women had to retrospectively recall their experience of adversity, which is different from current experience as they had to rely on the memory of which resources they had activated in the past. In addition to this, a subsequent study (Chew & Haase, 2016) disagreed on the number of factors that are present in the questionnaire.

6. Why is it important for a measure of family resilience to be disease specific? (paragraph 5). I’m also confused here that it states there is only one measure of family resilience in cancer (referencing the FRAS) but there is also the FRA.

We believe it is important to evaluate specific challenges that families are posed with when a member has cancer as it is an illness characterised by both chronic and acute phases in its trajectory. The same Walsh has adapted her model for illness and disabilities to show that there are specific aspects of family resilience related to diseases compared to other adverse events. The FRA is the only family resilience measure tested on oncological survivors, differently the FRAS was applied in two studies,

one recruited epileptic young adults, the other families with a child with ASD (autism spectrum disorder).

7. The authors criticise Sixbey's original development study but then follow this work by choosing, a priori to have six domains. Please justify this decision further.

We do follow Sixbey's study, choosing the same six factors of her work as we believed that Lane's three factors (that corresponded to the three overarching constructs of Walsh) did not capture all aspects of family resilience in dealing with an illness. Although Lane believes socio-economic resources had little value in BC survivors we believed that at least the social resources should be considered as a vital component of our measure due to the importance given to these in the current oncological settings. The limitations of Sixbey's measure are the reason for which we decided to construct a new, systemic measure of family resilience rather than validating Sixbey's individualistic questionnaire that was tested on the general American population. We agree that this was not clear in the Introduction and have modified it accordingly.

Materials and method

1. The second sentence of the opening paragraph is a bit confusing and needs amending slightly to spell out simply that the aim of study one was to develop a questionnaire (the Family Resilience Evaluation Questionnaire) to assess resilience in families affected by cancer.

Thank you, we have modified the text accordingly.

2. For Study 1, it would be helpful to have a couple of sentences about the overall plan – generate items through a literature review, carry out cognitive interviewing to check the items are comprehensible and then reduce the number of items.

The overall plan is now detailed at the beginning of the Methods section, under First study: Item generation phase and development of the questionnaire, page 8.

3. In order to demonstrate the content validity of a questionnaire, you need to show that it is representative of the entire construct it is meant to be measuring. There needs to be something more convincing than simply saying, "a thorough revision of scientific literature in the field of resilience and cancer" was carried out.

We agree and we regret that we did not go into the details of the procedure of our research study. Further details about the protocol of the study are now available in section 1.2.3.

4. The 'Face validity' section seems to me to describe more the process of cognitive interviewing to check comprehensibility: to check face validity you would need to check that people think the questionnaire is a measure of resilience. Please provide details of the ethics for this phase.

We agree with the reviewer and have removed the title "face validity" and described the process of cognitive interviewing under the title: First study: Item generation phase and development of the questionnaire. All study phases were presented in a single protocol to the local Ethics Committee, which was approved in all its parts on the 11th of April, 2016. We have added this sentence at the beginning of the Materials and Methods section.

5. The use of the word "proposed" with respect to inviting participants to take part in research needs to be amended. Also do not use the word "drop-out" to participants who withdrew from the project.

Thank you, we have changed the text accordingly.

6. What clinical information was collected about participants (section 1.2.1)? Where they recruited from outpatient clinics? Consecutive patients?

Clinical information about tumour stage, lymph nodes, and grading was retrieved from personal health records when available. With regards to the type of oncological therapy instead, the patient was asked with an open question whether they were currently under oncological treatment or had done oncological treatment in the past. If they replied affirmatively, they were asked which type of treatment. Within the informed consent, patients were asked if clinical data could be retrieved from the personal health records. Patients were recruited from two outpatient clinics within a comprehensive cancer hospital, the first clinic was of the Breast Cancer Division, the second of the Urology Division. Patients receiving oncological treatment were recruited from the Radiotherapy Division and Day Hospital. Compatibly with the researcher's availability, consecutive patients waiting for their appointment were enrolled in the study.

7. In section 1.2.2, it is stated that "The Italian version of the Resilience Scale for Adults (RSA) was chosen to assess the psychometric characteristics of the FaRE." I think you mean the concurrent validity.

We agree with this observation and we have changed the text accordingly.

8. What was the procedure (for participants)?

The procedure has now been added to section 1.2.3

9. What is meant by "cleanest" and most "interpretable" factor loading in section 1.2.4?

We agree that this wording of the sentence may be unclear. We meant that items were included in the factors were the ones with the highest loadings and those that belonged to a same psychological construct. We have modified the text accordingly.

10. What were the hypotheses for discriminative and divergent validity (what differences were you expecting to find and why)?

We have now detailed the hypothesis for construct validity (the correlations that we expected between RSA and FaRE) before conducting the analysis and the reasons why we were expecting positive correlations. This has been added at the end of page 16. The terms discriminative and divergent validity are not being used in the manuscript anymore.

Results

1. It would be good to see which items featured in which sub-scale in the supplementary material.

We have now included the items for each subscale under 1.2.6. We leave the questionnaire in the Appendix for reviewing purposes, however the copyright process of the questionnaire has begun, therefore it will not be possible to publish it openly online.

2. The issue for the section 1.2.5.5, relates back to point 10 above: why would you expect perceived social support to be different between breast and prostate patients, for example?

Differently from point 10, here we did not have any a priori expectations about differences between prostate and breast cancer patients.

3. Edit "adopted" version of questionnaire.

It has now been substituted with "final".

4. I have a problem with the religiousness and spirituality sub-scale. I am not surprised these items hang together well psychometrically but what do they tell us about family resilience? They just tell us whether a family is religious/spiritual or not.

Spirituality is one of the key factors of Walsh's family resilience framework and for this reason it was included in the questionnaire. It does not tell us only whether a family is religious or not, rather whether it is used as a resource to learn and grow from the adversity. Spirituality can allow families to have larger values and find a purpose in dealing with the illness. It is a resource in moments of distress which is often activated when facing cancer. It would be interesting to evaluate in future studies whether it is a transcultural resource.

5. Please provide some more details about the translation procedure and whether there were any cognitive interviews carried out with native English speakers. Some of the English is not very good e.g. items 1, 4.

We have now provided additional details about the translation procedure. Evaluation of the questionnaire was carried out with 10 healthy volunteers. We are sorry that we attached the previous English version, rather than the final one. We have now uploaded the final English version where the wording of some items was changed i.e. "deal" instead of "face" (item 12), "discuss" instead of "think about" (item 10), "important problems" instead of "significant difficulties" (item 9).

Study 2

1. This should follow the same overall structure as Study 1 (method, results)

The structure is now the same as Study 1 with regards to both the Method and Results section.

2. I would expect a validation study to provide some evidence that the questionnaire is measuring what it is intended to measure, rather than just validating the factorial structure, which is what the opening sentence suggests.

We agree with this remark and have added a description of what was expected of study 2, not only the validation of the factorial structure but also determining whether construct validity is present.

3. Again, I would like to know a bit more about how the participants were recruited and what clinical data were collected. What was the procedure for participants? Also, I was expecting some English participants

The procedure was the same as for the first study. Initially, the English partners of the European project were supposed to recruit patients in an English hospital, reason for which the English version of the questionnaire was created. However, due to organizational issues, this did not happen.

4. Why would you expect any differences between patients and caregivers (section 2.3.4)? I don't understand why there is a difference between perceived social support in section 2.3.4 but no difference in section 2.3.5

We decided to remove section 2.3.4 as a paired comparison between patient-caregiver dyad is a more accurate analysis of the differences between the dyad. Initially in 2.3.4 we had compared all patients against all caregivers.

5. I would lose section 2.3.7. These cut points need validating if they are going to be used as a screen for referral (and again, there is the issue of the religiousness and spirituality scale).

We agree, these cut-off points need validating and we have stressed this in the section 2.3.7, now 2.4.7. We have removed the subscale mean and standard deviations of the subscales and maintained the total family resilience score as this can be indicative of adjustment to illness, although it should be validated. We have addressed this limitation in the Discussion.

Discussion

1. The opening paragraph is too long. I suggest summarising the findings and then separating the other issues into different paragraphs. I am not sure why the final sentence is included.

We agree with this observation and have shortened the first paragraph and have separated the issues into different paragraphs. We have modified the final paragraph which provides conclusions to the paper.

2. Much of the rest of the discussion will need to be adapted in light of the other comments I have made above.

Thank you, we have adapted the discussion in light of the comments raised.

3. Future research would also need to provide evidence of test-retest reliability and responsiveness to change.

While we had already suggested the need for further research investigating test-retest reliability in our paragraph regarding the limitations of the study, we have now stressed this further.

Reviewer: 2

Reviewer Name: Inés Tomás

Institution and Country: University of Valencia, Spain

Please state any competing interests or state 'None declared': Non declared

Please leave your comments for the authors below

This paper aims to develop and validate an instrument to assess family resilience. The authors make sound of their contribution stating that there is only one published measure of family resilience in cancer patients (the Family Resilience Assessment scale, FRAS; Sixbey, 2005) that has been adequately validated. According to the authors, this measure has an individualistic view of resilience and does not capture the systemic processes involved. Then, a new measure, taking into account both the cancer patient's perspective and the caregiver's perspective, is developed. The authors develop two independent studies and provide evidence of reliability and validity of the scale. The starting point seem promising. However, there are some important theoretical and empirical issues that should be addressed.

- In the introduction section (page 4) the authors talk about Froma Walsh's work as a substantial contribution to research in family resilience. They also expose three different questionnaires that attempt to capture the multidimensional model of Walsh. However, they do not mention or name the 9 dimensions that this model initially proposed. I propose to the authors to start mentioning the 9 initial dimensions proposed in Walsh's model. That would be useful in order to state an initial starting point, and later talk about which dimensions were kept or not in following studies or development of the questionnaires. Furthermore, no specific information about the FRA is offered (number of items, subscales...).

We agree and we have now provided the names of the 9 dimensions of Walsh's framework. We hope this improved clarity of which dimensions have been removed or considered in the development studies. Information about FRA subscales and number of items has now been included.

- When presenting the aims of the two different studies carried out, I propose to the authors to consider the process of validation, which involves different stages: substantive, structural and external (Messick, 1995). The substantive stage defines and delineates the construct under investigation. The structural stage pertains to establishing evidence of factorial validity and reliability relative to the construct of interest. The external stage examines whether the construct under investigation is related to other variables in accordance with the theoretical expectations. Then, they should reformulate the aim of the studies according to this framework, as the substantive stage seem not to be mentioned.

We have read Messick's validation process for psychological measures and have stressed this more clearly at the end of the Introduction and at the beginning of the Material and Methods section at page 7. We have expanded on the substantive stage at the end of page 7.

- The authors should state more clearly what are the contributions of the new created measure in regard to the previous one (the FRAS). It seems that the new measure has been created following the same 6 factor structure developed and validated for the FRAS. Why then create this new measure? What for? What are the specific contributions of the FaRe regarding the FRAS? Some point has been made by the authors regarding this issue, when they state that the FRAS has an individualistic view of resilience. However, could it just be solved by changing the initial instructions sentence? This point is not clear. More justification should be provided in order to justify to contribution of developing this new measure.

The FRAS does not only have an individualistic view, items of the questionnaire were dropped when the measure was administered to the normal population which had not necessarily experienced an adversity. Walsh herself adapted her family resilience framework to families with illness and disability in order to show that some of the constructs may vary depending on the context. The current measure (Fare) is illness-specific, the preliminary measure with 60 items wanted to investigate which family resilience aspects are salient in an oncological population as these might differ from the normal population (sample of the FRAS). Hopefully, future studies will test it on other illnesses too and evaluate whether these aspects are salient for both patients and caregivers in different medical settings.

We think that it cannot be solved by changing the initial instruction sentence because the wording of the statements differs as they have been constructed with the help of systemic psychotherapists that enclosed family processes rather than how the individuals see themselves compared to their family.

- Developing a questionnaire implies to accurately define the construct we want to measure, to deeply analyze and expose the model that we use to define the construct, and to discuss on the specific dimensions that configure the construct. None or poor development on that has been presented in the manuscript. In the introduction section I propose to the authors to further develop on that: what is resilience? What is family resilience? Which are the different models, theories or approaches on resilience/ family resilience? Which one has been selected and why? What does this model states about resilience/family resilience? Which are the dimensions that are considered in the model? What does each one mean or represent?

Dear Reviewer, a detailed analysis of what is resilience, family resilience, the different models and theories to family resilience has been presented in our previous work: Faccio F, Renzi C, Giudice A V., Pravettoni G. Family Resilience in the Oncology Setting: Development of an Integrative Framework. *Front Psychol.* 2018;9(666). doi:10.3389/fpsyg.2018.00666. We thought it was beyond the scope of the paper to review again these aspects. However, we have expanded a little on Walsh's family resilience model, its dimensions and why it was chosen compared to previous family resilience models. If this is not enough, we are open to expanding further.

- In page 8, the authors indicate that "The Italian version of the Resilience Scale for Adults (RSA) was chosen to assess the psychometric characteristics of the FaRE". I suggest to be more concrete

on that sentence and substitute “to assess the psychometric characteristics of the FaRe” for a more specific one (e.g., “to assess the convergent validity of the FaRe”; or “to provide evidence of validity of the FaRe based on the relationship with other variables” (following the current APA guidelines regarding validation of scales)).

This sentence has now been modified, thank you.

- In page 8, a established scale (the Resilience Scale for Adults, RSA; Capanna et al., 2015) for measuring resilience is introduced to be used in the validation process of the FaRe. The authors indicate that this scale is composed of six resilience factors: four factors measure the individual’s characteristics, one the family environment, and the last one social networks. It seems that two of this dimensions (family environment and social network) represent family resilience. Again, more justification on the need of developing a new measure of family resilience, that do not overlap with previous published and well established scales, is needed.

The RSA is a well-known measure of individual resilience and yes, within its subscales we can find Family Cohesion and Social Resources, which are only two of the six subscales and the same RSA does not aim to measure family resilience, rather the individual’s resilience. In addition to this, the Family Cohesion RSA subscale has a lot of different aspects, such as communication, mutual appreciation and conflict resolution, but does not include family coping strategies. RSA Social Resources instead concerns how the individual looks for social support, rather than the type of social resources that the family can put in place. While there is some overlap between the two subscales, the RSA does not measure and did not aim to measure family resilience. It was used to assess whether there was convergent validity on some subscales.

- Nothing about data collection procedure is stated in the manuscript.

We have now included the procedure under section 1.2.3

- In the Statistical analyses section, the authors rightly state that the first step was to carry out an EFA. I agree that the authors select the adequate statistical technique. However, the extraction method (principal axis factoring) and the rotation method (varimax rotation) selected to carry out the EFA are not the recommended ones.

It is also stated that “factors with eigenvalues greater than 1.0 were retained”. Although Kaiser’s rule is still the most widely used criteria to select the number of factors to retain, it is also the less recommended of all possible options.

I would suggest the authors to review the revised and updated guidelines to carry out an EFA (Lloret-Segura, Ferreres-Traver, Hernández-Baeza, & Tomás-Marco, 2014).

Moreover, there is no information about the statistical package used to carry out the analysis.

We have read carefully the updated guidelines suggested by the reviewer. We re-ran the EFA using orthomax rotation as a rotation method and maximum likelihood as an extraction method; the results were the same as those encountered with the previous methods. We have added information about the statistical package, SAS version 9.2, under section 1.2.4.

- Description of the sample is offered in the Results section. This information should appear in the Method section.

We have described the sample characteristics in the Results section as descriptive statistics were conducted on the data, for this reason the table and the other percentages are represented in the Results section. In the Method section we kept inclusion and exclusion criteria and sample size.

- Regarding the EFA results, no information on the adequacy of data to be factor analyzed is provided (e.g., Kaiser Meyer Olkin's KMO test for sampling adequacy). Moreover, I would not completely rely on the results of the EFA, considering that the options to carry out this analysis were not the most adequate ones (as previously stated). Furthermore, the scree plot that appears in page 24 seem to indicate a six factor solution as a plausible one.

More information on the criteria used to take decisions should be offered. Why the "Ability to make meaning from adversity" factor was removed?

Regarding the information provided of goodness of fit indices, it is not clear whether this fit correspond to the initial model with the complete pool of items, or to the model with the removed items. Moreover, information of the number of items that were developed for the initial 6 dimensions of the questionnaire, and number of items removed should be offered....

We have now conducted KMO test for sampling adequacy: overall KMO=0.92, clinical sample KMO=0.9, caregivers KMO=0.84. All values above 0.7 indicate that the sampling is adequate. These values have now been inserted in the manuscript under 1.2.5.2.

CFI is already a good measure for the adequacy of data and it is the only statistical analysis available in SAS. In addition to this, the scree plot was used. While reading of scree plots can be subjective, the elbow of the curve and the percentage of variance of factor 5 and 6 are so small that they do not add much to the whole model. In Social Sciences, 50% or above of variance is enough, ours is over 60%.

We have stated that the questionnaire was initially composed of 60 items; 36 of them were removed as they did not "hang" together. Factor 5, The Ability to make meaning of adversity was removed because it was composed of 2 items that were unrelated to each other. Factor 6 was composed of 3 items that once again did not have a common underpinning construct, for this reason it was removed. The goodness of fit was run in both Study 1 and 2 and we re-ran them after changing the rotation and extraction method. The CFI of Study 1 was the same, while for Study 2 it was 0.9 instead of 0.88.

- I have a concern regarding the final questionnaire composition. The initial pool of items was 60, and before the EFA, just 24 were retained. This item deletion could have affected the construct representation? That is, some specific and valuable aspects could now not be represented by the items remaining in the questionnaire? Moreover, 6 dimensions were proposed initially, and finally just 4 were kept. This dimension dropping could be affecting the construct validity? That is, could be that some relevant dimension for family resilience is now missing?

As stated above, we believe that construct validity is not affected as the psychological aspects of family resilience are represented in the questionnaire. Only 24 items were retained as the others were not present in the factor loadings. This might also suggest that family resilience could have slightly different aspects in illness management, particularly in cancer care.

- To further assess the internal reliability of the scales, the authors should also provide the Average Variance Extracted (AVE) values and composite reliability values (ρ).

SAS software does not run AVE. While composite reliability values are indeed a good alternative to Chronbach's alpha we follow Peterson, Robert and Yolib Kim (2013) in stating that the difference between the use of the two methods is inconsequential. See Peterson, Robert A., and Yeolib Kim. "On the relationship between coefficient alpha and composite reliability." *Journal of Applied Psychology* 98.1 (2013): 194.

- Authors should pay attention to which pieces of information appear in which section. For example, include in the Analyses section all the information related to the analyses carried out, and leave the Results section just to comment on specific results. As an example, the following text figures in the Results section, when it should be placed in the Analyses section : "In order to assess

construct validity of FaRE, we analyzed whether the subscales of the questionnaire correlated with the RSA using Spearman correlations”.

This text has now been moved to the Analysis section, thank you. We have also moved other information related to the analyses in the specific section.

- Authors are also encouraged to address the validation process within the framework of the AERA, APA and NCME (2014) guidelines. It affects the expressions and terms used, as it is recommended to refer to evidences of validity and the different sources of evidence of validity. [American Educational Research Association, American Psychological Association, & National Council on Measurement in Education. (1999). Standards for educational and psychological testing. American Educational Research Association.]

Thank you, we have reviewed these guidelines and Messick's (1994) steps to validate a psychological assessment and have changes the terms and expressions accordingly (Messick reference: Messick S. VALIDITY OF PSYCHOLOGICAL ASSESSMENT: VALIDATION OF INFERENCES FROM PERSONS' RESPONSES AND PERFORMANCES AS SCIENTIFIC INQUIRY INTO SCORE MEANING. ETS Res Rep Ser. 1994;1994(2):i-28. Doi:10.1002/j.2333-8504.1994.tb01618.x.)

- In order to provide evidence of validity of test scores of the FaRe based on the relationship with other variables, as previously indicated, the authors also collect measures with the Resilience Scale for Adults (RSA). However, it should be advised which kind of relationships are expected in order to confirm that the results provide evidence of validity. Which kind of pattern relationship are expected? Which dimensions should show higher relationships according to their content and meaning? Are these expected pattern in the correlations supported? We also encourage the authors to present the correlations in a table for better understanding), and justify why they offer Spearman correlations instead of Pearson correlations.

We have now detailed the correlations that we expected between RSA and FaRE before conducting the analysis and the reasons why we were expecting positive correlations. This has been added at page 16.

Our data was not normally distributed, and this has occurred in previous studies using the RSA; therefore, we chose a non-parametric correlation test, such as Spearman. We have added the justification at page 17.

We have maintained only values of construct validity for the full sample of breast and prostate cancer patients.

- I cannot see the point of the results that appear in the “1.2.5.5. Discriminant validity” section. In which way the reported results provide evidence of discriminant validity? Are there any previous theoretical or empirical foundations to expect that the values in the dimensions of the FaRe scale should be different among patients and caregivers, or among breast and prostate cancer patients?

We agree that this is not discriminant validity, we have removed the title and maintained the comparisons between both groups as we believe they are interesting preliminary results that can inform future research and clinical work. However, if the reviewers believe that the differences in scores should not be shown we are open to removing it.

- In the second study, no information on the kind of factorial analysis that was carried out is reported. Was is a Confirmatory Factor Analysis (CFA)? Following the recommendation on scale validation, in this step, a CFA should be used in order to confirm the factorial structure that appeared in the EFA carried out in Study 1. Again, no information on the statistical software used is provided.

We agree, we are sorry that we forgot to mention that CFA was conducted, we have now added this and the statistical package.

- In the Discussion section it is stated: “The questionnaire confirms Sixbey’s assumption that the nine-factor family resilience model developed by Walsh can be reduced”. To state that, the FaRe questionnaire developed in this study should have considered the initial 9 factor proposed in Walsh’s model. However, the starting point was a 6 factor model.

As we modified the Introduction, the Discussion underwent significant changes with regards to the comments of the other questionnaires compared to the FaRE. We are sorry for not having explained clearly the starting model, which was indeed Sixbey’s 6 factor model. We have changed the text accordingly at the beginning of page 21, thank you.

- In the Discussion section authors also state: “Lane’s family resilience measure had highlighted that social and economic resources had little value to breast cancer survivors and our measure confirms this”. From which part of the results section can this conclusion be drawn?

We are sorry for not having explained clearly the similarity and difference with Lane’s construct. Initially, our questionnaire had the factor Social and Economic resources, however after conducting factorial analysis only the social resources (“Perceived social support”) were related together in one factor, therefore the economic aspects were dropped and not considered together with the social ones. We have changed the meaning of this sentence, which can now be found in page 21, thank you.

- Most of the discussion section focus on the comparison between breast and prostate cancer patients scores, and between patients and care givers scores. None of this comparisons are in the aim of the study.

We agree with this observation and have reduced significantly the comparisons and have linked them to one of our limitations, the low inter-rater reliability between patients and caregivers (page 22), thank you.

Reviewer: 3

Reviewer Name: Erik Farin-Glattacker

Institution and Country: Section of Health Care Research and Rehabilitation Research, Faculty of Medicine and Medical Center - University of Freiburg, Germany

Please state any competing interests or state ‘None declared’: None declared

Please leave your comments for the authors below

The manuscript reports on the development and validation of a family resilience questionnaire. This is a relevant research topic and the study seems carefully conducted, but there are some methodological vagueness and weaknesses.

My major concern is about the handling of patient and caregiver data. It seems as if these data were combined to conduct psychometric testing with the whole data set. As the perspectives of patients and caregivers concerning adaptation flexibility to the disease of the patient may be quite different, this approach seems disputable. I would advise to test first factorial invariance of the FaRE instrument in patients and caregivers. There are at least two levels of factorial variance that may be of interest in this case: „weak factorial invariance” across patients and caregivers and “strong factorial invariance”.

Comparing mean values of patients and caregivers in a methodological paper without testing factorial invariance across these groups may lead to false conclusions.

We agree and we have now conducted weak and strong factorial invariance across patients and caregivers using MCFA (Multigroup Confirmatory Factor Analysis). Strong factorial invariance was not supported, while weak factorial invariance was supported (patients SRMR = 0.079, caregivers SRMR = 0.080). According to Kang et al. (2016) "achieving weak factorial invariance is considered sufficient to proceed with group comparisons..." (Byrne, Shavelson, & Muthén, 1989; Horn, McArdle, & Mason, 1983)." This has now been added under 1.2.5.2.

Concerning chapter 1.2.5.5.1 Comparison between patients and caregivers: As patients and caregivers are approached as raters of the same concept (family resilience) I would advise to compute intraclass correlations (ICC) as estimates of interrater reliability.

We have computed ICC for all factors and the coefficients were between 0.44 and 0.53, indicating overall weak to moderate interrater reliability, suggesting that there is some variation in the rating of the same construct depending on the sample group. We have now included this in the manuscript under the comparison between patients and caregivers and addressed it in the limitations of the study.

Testing construct validity demands clear hypotheses about the expected correlations (e.g., COSMIN checklist manual). These hypotheses should be stated and justified. In the current version of the manuscript this is missing. Factors that were considered in construct validity testing differed in the methods section (chap. 1.2.4) and in the results section (chap. 1.2.5.4).

We have now stated and justified the hypothesis for construct validity. We have checked the COSMIN checklist for construct validity, unfortunately we were not able to test cross-cultural validity as the recruitment in an English hospital was not conducted due to unforeseen reasons.

The factors considered in construct validity testing are the same in the two sections.

The kind of the relation between patient and caregiver (e.g., son/daughter, wife/husband) was ignored. At least the authors should give this information in the table depicting characteristics of the sample.

Sociodemographic characteristics of the caregivers samples are depicted in Table 1, we have added relation between patient and caregiver as percentages in text under 1.2.5.1.

The criteria that lead the authors to extract 4 factors in the factor analysis should be stated more clearly. In my view the scree plot does not support this. Were aspects of interpretability decisive?

While reading of scree plots can be subjective the elbow of the curve and the percentage of variance of factor 5 and 6 are so small that they do not add much to the whole model. In Social Sciences, 50% or above of variance is enough, ours is over 60%. In addition to this, interpretability of the items of Factor 5 and Factor 6, which did not hang together, pushed towards a 4 factor solution.

As a rule of thumb, 5 participants per item are necessary to conduct exploratory factor analyses (60 items * 5 participants = 300). In view of this, the sample size of the two sub-studies is too small. At least, this point should be discussed as limitation.

We agree with regards to the first study, for the second study the number of items was 24 (24*5=120) and we recruited a higher sample than the recommended size. We will discuss this in the limitations section, thank you.

The inclusion criteria pose some questions: What is the reason for different age inclusion criteria for patients (>=25, <=80) and caregivers (>=18)?

In our comprehensive cancer centre, breast cancer patients under 25 years old are extremely rare and we believed that they were not representative of the breast cancer population (there are no under 25 year olds in the prostate cancer population).

The age inclusion criteria for caregivers was lower than the patients because we wanted to include young adults who accompanied their parents to consultations. This said the youngest caregiver was 26 years old, so no one was in the age range 18-25 years old in both groups.

Page 14: „The Bentler Comparative Fit Index was 0.88, the RMR Estimate 0.07, indicating a good model of fit between the model and the data.“ I can't follow this. On page 10 the authors state that CFI should be greater than 0.90.

We agree that good fit is a strong word for the CFI of 0.88. We changed rotation and extraction method as the first reviewer suggested and have updated the results. CFI is now 0.9 and we consider therefore an adequate fit between model and data.

Page 16: „The FaRE questionnaire is designed to measure family resilience in medical settings.“ This assertion seems to be overdrawn as the authors investigate only cancer patients.

We agree with this and have substituted it with “oncological settings”. Future studies should evaluate whether these aspects of family resilience measured with the FaRE show the same pattern of responses in other chronic illnesses.

The discussion should include a more thorough consideration of limitations. The strengths of the study mentioned in the discussion section („The sample size, the use of a standardised measure to compare factors, the goodness of fit between model and data, and the high internal consistency are some of the strengths of the study.“) should be reconsidered. The sample size is quite small given the statistical analyses that were conducted. Goodness of fit values and internal consistency values are results, not strengths.

We agree with this observation and we have given more consideration to the limitations of the study, including small sample size for the first study.

VERSION 2 – REVIEW

REVIEWER	Sally Wheelwright University of Southampton, UK
REVIEW RETURNED	19-Oct-2018

GENERAL COMMENTS	I think the authors have done a very good job at addressing the previous comments and the paper is much improved. However, I do not agree with the response to my concerns about the religiousness and spirituality sub-scale. My concern is that this scale simply assesses whether a family is religious or not. The authors argues that “...[spirituality} is used as a resource to learn and grow from the adversity. Spirituality can allow families to have larger values and find a purpose in dealing with the illness. It is a resource in moments of distress which is often activated when facing cancer.” I think my criticism stands when you actually look at the content of the items in this scale. There are 4 items which ask whether you attend places of worship, believe in a supreme spiritual being, participate in the activities of the religious
--

	community and ask for advice from a religious/spiritual reference figure. At the very least, this is a limitation to the study. Alternatively, the authors may want to consider a total score which does not include this scale. The only other suggestion I have is that the information about the English version is not required as this is not used in the study.
--	---

REVIEWER	Inés Tomás University of Valencia (Spain)
REVIEW RETURNED	18-Dec-2018

GENERAL COMMENTS	I am suitably impressed by the authors' efforts to manage feedback offered in the previous revision. I find that the revised manuscript is improved and congratulate the authors to this end. My impression is that authors have been responsive to most of the issues I have raised in my review of the original submission. Few issues remain from my point of view. I have tried to clarify these, including other issues related to the revised manuscript.  • In my previous review I indicated: "To further assess the internal reliability of the scales, the authors should also provide the Average Variance Extracted (AVE) values and composite reliability values (rho)." Authors stated that: "SAS software does not run AVE. While composite reliability values are indeed a good alternative to Chronbach's alpha we follow Peterson, Robert and Yolib Kim (2013) in stating that the difference between the use of the two methods is inconsequential. See Peterson, Robert A., and Yeolib Kim. "On the relationship between coefficient alpha and composite reliability." Journal of Applied Psychology 98.1 (2013): 194." I was not proposing to use other reliability indices as an alternative to Cronbach's alpha, but to offer additional measures of reliability in addition to Cronbach's alpha. AVE and rho values are easily computed just with the information of the factor loadings obtained from the factor analysis. AVE values are computed as the sum of squared standardized factor loadings divided by the sum of squared standardized factor loadings plus the sum of error variances. A value of .50 or greater indicates a good score reliability as the variance of the construct is greater than the error variance (Fornell & Larcker, 1981). Composite reliability values (rho) are computed as follow: $(\text{sum of standardized loadings})^2 / \{(\text{sum of standardized loadings})^2 + (\text{sum of error variances})\}$. A value of .70 or greater indicates an acceptable reliability (Raykov, 2001). It is quite easy to just implement the formulas in an Excel document and compute the values of the reliability measures just using the factor loading provided from the factor analysis. Nevertheless, an Excel sheet with the implementation of the formulas of AVE and Rho are provided to the authors to simplify the computation. Fornell, C., & Larcker, D. F. (1981). Evaluating structural equation models with unobservable variables and measurement error. Journal of Marketing Research, 18, 39-50. Raykov, T. (2001). Estimation of congeneric scale reliability using covariance structure analysis with nonlinear constraints. British Journal of Mathematical and Statistical Psychology, 54, 315-323.
--

• In response to a comment from Reviewer 3, in the revised version of the manuscript, authors have conducted multigroup factor analysis to test for weak and strong factorial invariance across patients and caregivers. Authors conclude that strong factorial invariance was not supported, while weak factorial invariance suggested acceptable measurement invariance (patients SRMR =0.079, caregivers SRMR= 0.080). Moreover, they state that weak factorial invariance is enough to proceed with group comparisons.

First of all, I do not agree with the statement that achieving weak factorial invariance is sufficient to proceed with group comparisons. According to Meredith (1993), under appropriate conditions, a particular item would be strongly factorial invariant when both the factor loading and the intercept are invariant across the subpopulations derived by selection on a grouping variable. If strong factorial invariance holds, between-group differences in average score of an item will reflect between-group differences in latent means. Therefore, when strong factorial invariance is supported, average item and scale scores are comparable across groups.

In addition, I do not also understand why the authors conclude that strong factorial invariance was not supported. The goodness of fit of the alternative models should be compared (e.g., chi-square difference tests or differences in practical fit indices can be used). For example, to interpret differences in practical fit indices, support for the more parsimonious model requires a change in CFI of less than .01 (Chen, 2007; Cheung & Rensvold, 2001) or a change in RMSEA of less than .015 (Chen, 2007).

Moreover, if multigroup CFA is going to be carried out in Study 1, it should be indicated in the Statistical Analyses section of Study 1, and also indicate the purpose of this analysis. It is important to notice that testing factorial invariance across patients and caregivers, is a guarantee to avoid leading to false conclusions when comparing mean values of patients and caregiver. In the Results section of Study 1, information about EFA and multigroup CFA appears mixed up. The results of the EFA should appear first, and then, once the factorial structure of the questionnaire is established, results of the multigroup CFA (carried out with the final version of the questionnaire) should appear.

Chen, F. F. (2007). Sensitivity of goodness of fit indexes to lack of measurement invariance. *Structural Equation Modeling*, 14, 464-504. <https://doi.org/10.1080/10705510701301834>.

Cheung, G. W., & Rensvold, R. B. (2002). Evaluating goodness-of-fit indexes for testing measurement invariance. *Structural Equation Modeling*, 9(2), 233-255. https://doi.org/10.1207/s15328007sem0902_5.

Meredith, W. (1993). Measurement invariance, factor analysis and factorial invariance. *Psychometrika*, 58(4), 525–543.

• Results of CFA in study 2 and of multigroup CFA in study 1 should include information about the chi-square and degrees of freedom of the tested models. Moreover, the reported goodness of fit indices should be the same along the different CFA carried out in the manuscript. Information about the range of values of the

	factor loadings and if they were statistically significant should also be reported. The reviewer provided a marked copy with additional comments. Please contact the publisher for full details.
--	--

VERSION 2 – AUTHOR RESPONSE

Reviewer: 2

Reviewer Name: Inés Tomás

Institution and Country: University of Valencia (Spain)

Please state any competing interests or state 'None declared': None declared

Please leave your comments for the authors below

I am suitably impressed by the authors' efforts to manage feedback offered in the previous revision. I find that the revised manuscript is improved and congratulate the authors to this end.

My impression is that authors have been responsive to most of the issues I have raised in my review of the original submission. Few issues remain from my point of view. I have tried to clarify these, including other issues related to the revised manuscript.

- In my previous review I indicated: "To further assess the internal reliability of the scales, the authors should also provide the Average Variance Extracted (AVE) values and composite reliability values (rho)."

Authors stated that: "SAS software does not run AVE. While composite reliability values are indeed a good alternative to Chronbach's alpha we follow Peterson, Robert and Yolib Kim (2013) in stating that the difference between the use of the two methods is inconsequential. See Peterson, Robert A., and Yeolib Kim. "On the relationship between coefficient alpha and composite reliability." *Journal of Applied Psychology* 98.1 (2013): 194."

I was not proposing to use other reliability indices as an alternative to Cronbach's alpha, but to offer additional measures of reliability in addition to Cronbach's alpha. AVE and rho values are easily computed just with the information of the factor loadings obtained from the factor analysis. AVE values are computed as the sum of squared standardized factor loadings divided by the sum of squared standardized factor loadings plus the sum of error variances. A value of .50 or greater indicates a good score reliability as the variance of the construct is greater than the error variance (Fornell & Larcker, 1981). Composite reliability values (rho) are computed as follow: $(\text{sum of standardized loadings})^2 / \{(\text{sum of standardized loadings})^2 + (\text{sum of error variances})\}$. A value of .70 or greater indicates an acceptable reliability (Raykov, 2001). It is quite easy to just implement the formulas in an Excel document and compute the values of the reliability measures just using the factor loading provided from the factor analysis. Nevertheless, an Excel sheet with the implementation of the formulas of AVE and Rho are provided to the authors to simplify the computation.

Fornell, C., & Larcker, D. F. (1981). Evaluating structural equation models with unobservable variables and measurement error. *Journal of Marketing Research*, 18, 39-50.

Raykov, T. (2001). Estimation of congeneric scale reliability using covariance structure analysis with nonlinear constraints. *British Journal of Mathematical and Statistical Psychology*, 54, 315-323.

Dear Reviewer, thank you for your comments. We have used the file you sent us as a guide and inserted the formulas in Excel to calculate AVE values and rho scores on the final version of the questionnaire. These values have now been added at page 13, and they confirm good reliability of the scale.

- In response to a comment from Reviewer 3, in the revised version of the manuscript, authors have conducted multigroup factor analysis to test for weak and strong factorial invariance across patients and caregivers. Authors conclude that strong factorial invariance was not supported, while weak factorial invariance suggested acceptable measurement invariance (patients SRMR =0.079, caregivers SRMR= 0.080). Moreover, they state that weak factorial invariance is enough to proceed with group comparisons.

First of all, I do not agree with the statement that achieving weak factorial invariance is sufficient to proceed with group comparisons. According to Meredith (1993), under appropriate conditions, a particular item would be strongly factorial invariant when both the factor loading and the intercept are invariant across the subpopulations derived by selection on a grouping variable. If strong factorial invariance holds, between-group differences in average score of an item will reflect between-group differences in latent means. Therefore, when strong factorial invariance is supported, average item and scale scores are comparable across groups.

In addition, I do not also understand why the authors conclude that strong factorial invariance was not supported. The goodness of fit of the alternative models should be compared (e.g., chi-square difference tests or differences in practical fit indices can be used). For example, to interpret differences in practical fit indices, support for the more parsimonious model requires a change in CFI of less than .01 (Chen, 2007; Cheung & Rensvold, 2001) or a change in RMSEA of less than .015 (Chen, 2007).

Moreover, if multigroup CFA is going to be carried out in Study 1, it should be indicated in the Statistical Analyses section of Study 1, and also indicate the purpose of this analysis. It is important to notice that testing factorial invariance across patients and caregivers, is a guarantee to avoid leading to false conclusions when comparing mean values of patients and caregiver. In the Results section of Study 1, information about EFA and multigroup CFA appears mixed up. The results of the EFA should appear first, and then, once the factorial structure of the questionnaire is established, results of the multigroup CFA (carried out with the final version of the questionnaire) should appear.

Chen, F. F. (2007). Sensitivity of goodness of fit indexes to lack of measurement invariance. *Structural Equation Modeling*, 14, 464-504. <https://doi.org/10.1080/10705510701301834>.

Cheung, G. W., & Rensvold, R. B. (2002). Evaluating goodness-of-fit indexes for testing measurement invariance. *Structural Equation Modeling*, 9(2), 233-255. https://doi.org/10.1207/s15328007sem0902_5.

Meredith, W. (1993). Measurement invariance, factor analysis and factorial invariance. *Psychometrika*, 58(4), 525–543.

Dear Reviewer, thank you for your comment. We have tested for a more parsimonious model looking at RMSEA change between the two models and the change was less than .015. For this reason, we have now stated that strong factorial invariance is supported, allowing to proceed with group comparisons. We have stated factorial invariance as one of the statistical analyses conducted in Study 2 at the bottom of page 15 and we have reported the exact values of RMSEA change under 2.4.2 at page 17. We have corrected the order in which results appear, EFA in study 1, CFA in Study 2.

- Results of CFA in study 2 and of multigroup CFA in study 1 should include information about the chi-square and degrees of freedom of the tested models. Moreover, the reported goodness of fit indices

should be the same along the different CFA carried out in the manuscript. Information about the range of values of the factor loadings and if they were statistically significant should also be reported.

Thank you very much for your comment. Chi square results have now been reported in both study 1 and study 2 for the factorial analyses that have been conducted. We have now reported the goodness of fit indices in the same manner across the manuscript. We have reported values of factor loadings and difference in the scores in the table.

VERSION 3 - REVIEW

REVIEWER	Inés Tomás Universitat de València (Spain)
REVIEW RETURNED	28-Jan-2019

GENERAL COMMENTS	I think the authors have done a very good job at addressing the previous comments. However, there is still one issue that remains unclear.  • In response to a comment from Reviewer 3, in the revised version of the manuscript, authors included a multigroup factor analysis to test for weak and strong factorial invariance across patients and caregivers. I think that authors do not succeed to clearly explain this analysis. They do not mention which models are compared, and do not report the goodness of fit of the alternative models in order to make clear the model comparison. My recommendation to the authors would be to explain this analysis more clearly. I also suggest following this steps:  1) Testing the baseline model separately in each of the groups (e.g., patients and caregivers). 2) Testing the structural invariance multigroup model: all parameters are freely estimated. 3) Testing the weak measurement invariance multigroup model: the factor loadings are fixed to be invariant across groups. 4) Testing the strong factorial invariance multigroup model: factor loadings and intercepts are fixed to be invariant across groups. The goodness of fit of the alternative models should be reported and compared (e.g., chi-square difference tests or differences in practical fit indices can be used). For example, to interpret differences in practical fit indices, support for the more parsimonious model requires a change in CFI of less than .01 (Chen, 2007; Cheung & Rensvold, 2001) or a change in RMSEA of less than .015 (Chen, 2007). As the strong factorial invariance model is the more parsimonious (the one with less parameters), no significant or practical differences in goodness of fit indices would provide evidence for strong factorial invariance. Remember that if strong factorial invariance holds, between-group differences in average score of an item will reflect between-group differences in latent means. Therefore, when strong factorial invariance is supported, average item and scale scores are comparable across groups. Considering the reported information in page 16 (2.4.2. Confirmatory factor analysis), authors report the fit of a model, but they do not clarify which model this fit belongs to. Authors do not
--

	report the complete information of the goodness of fit indices of the weak and strong factorial invariance models.
--	--

VERSION 3 – AUTHOR RESPONSE

In response to a comment from Reviewer 3, in the revised version of the manuscript, authors included a multigroup factor analysis to test for weak and strong factorial invariance across patients and caregivers. I think that authors do not succeed to clearly explain this analysis. They do not mention which models are compared, and do not report the goodness of fit of the alternative models in order to make clear the model comparison. My recommendation to the authors would be to explain this analysis more clearly. I also suggest following this steps:

- 1) Testing the baseline model separately in each of the groups (e.g., patients and caregivers).
- 2) Testing the structural invariance multigroup model: all parameters are freely estimated.
- 3) Testing the weak measurement invariance multigroup model: the factor loadings are fixed to be invariant across groups.
- 4) Testing the strong factorial invariance multigroup model: factor loadings and intercepts are fixed to be invariant across groups.

The goodness of fit of the alternative models should be reported and compared (e.g., chi-square difference tests or differences in practical fit indices can be used). For example, to interpret differences in practical fit indices, support for the more parsimonious model requires a change in CFI of less than .01 (Chen, 2007; Cheung & Rensvold, 2001) or a change in RMSEA of less than .015 (Chen, 2007). As the strong factorial invariance model is the more parsimonious (the one with less parameters), no significant or practical differences in goodness of fit indices would provide evidence for strong factorial invariance.

Remember that if strong factorial invariance holds, between-group differences in average score of an item will reflect between-group differences in latent means. Therefore, when strong factorial invariance is supported, average item and scale scores are comparable across groups.

Considering the reported information in page 16 (2.4.2. Confirmatory factor analysis), authors report the fit of a model, but they do not clarify which model this fit belongs to. Authors do not report the complete information of the goodness of fit indices of the weak and strong factorial invariance models.

Thank you for your comment. We had already tested the model separately for patients and caregivers at the end of page 16, and SMSR values for both groups were reported; here weak measurement invariance was tested by maintaining the factor loadings invariant across groups. We also tested the strong factorial invariance multigroup model, where factor loadings and intercepts were fixed to be invariant across groups. We followed your indication and also the one offered by the Assistant Editor to report explicitly the goodness of fit of both models to make the comparison as clear and as transparent as possible. As strong factorial invariance was supported, we have removed comparison between patients/caregivers in average scale scores in study 2; as suggested, if strong factorial invariance is supported, the scores they are comparable across groups.

We have now explicitly reported complete information of goodness of fit indices of weak and strong factorial invariance models under 2.4.2. Confirmatory factor analysis at page 17. We have also added an explanation of the procedure at the bottom of page 15 under 2.3 Statistical analyses. We have also expanded further on the analysis conducted in Study 1 under 1.2.4 to state clearly all the procedures

we conducted to ensure model adequacy and fit. We have added references to the AVE and Rho score calculations.

We kindly ask you to now consider the manuscript for publication.